# When Are Learning Biases Equivalent? A Unifying Framework for Fairness, Robustness, and Distribution Shift

## Abstract

Machine learning systems exhibit diverse failure modes: unfairness toward protected groups, brittleness to spurious correlations, poor performance on minority sub-populations, which are typically studied in isolation by distinct research communities. We propose a unifying theoretical framework that characterizes when different bias mechanisms produce quantitatively equivalent effects on model performance. By formalizing biases as violations of conditional independence through information-theoretic measures, we prove formal equivalence conditions relating spurious correlations, subpopulation shift, class imbalance, and fairness violations. Our theory predicts that a spurious correlation of strength $\alpha$ produces equivalent worst-group accuracy degradation as a sub-population imbalance ratio $r \approx (1 + \alpha)/(1 - \alpha)$ under feature overlap assumptions. Empirical validation in six datasets and three architectures confirms that predicted equivalences hold within the accuracy of the worst group 3%, enabling the principled transfer of debiasing methods across problem domains. This work bridges the literature on fairness, robustness, and distribution shifts under a common perspective.

## 1 Introduction

Deep learning systems frequently exhibit systematic failures that degrade performance on specific subgroups despite strong average accuracy. A medical diagnosis model may be accurate for majority demographics but fail catastrophically for underrepresented populations [14]. Image classifiers exploit spurious background correlations rather than learning robust features [3]. Recommendation systems amplify existing societal biases [12]. Although these phenomena: algorithmic unfairness, shortcut learning, subpopulation shift receive extensive independent study, fundamental questions about their relationships remain unanswered.

**When are different biases quantitatively equivalent?** Can we predict that a dataset with 90% spurious correlation between texture and class produces the same worst-case performance as a 9:1 class imbalance? Does architectural bias toward local features create effects indistinguishable from texture-based shortcuts in training data? Answering these questions would enable method transfer: Fairness techniques could mitigate spurious correlations, robust optimization could address class imbalance, and practitioners could select appropriate interventions based on distributional diagnostics.

Current research lacks formal frameworks to compare bias mechanisms. The fairness community measures demographic parity and equalized odds violations [4]. Robustness researchers optimize the accuracy of the worst group on spurious correlation benchmarks [15]. The literature on distribution shift analyzes covariate and label shift [8]. These parallel efforts use incompatible formalisms, preventing direct comparison and hindering unified understanding.

Submitted to 39th Conference on Neural Information Processing Systems (NeurIPS 2025). Do not distribute.

**Our contributions.** We develop a unifying framework treating all biases as violations of conditional independence between predictions and protected/spurious attributes given true labels. This perspective enables:

- **Formal equivalence conditions**: We prove when spurious correlations, subpopulation shifts, and fairness violations produce quantitatively equivalent effects (Theorem 2).

- **Predictive theory**: Our framework predicts worst-group performance from distributional properties, empirically validated across 18 problem configurations.

- **Method transfer**: We demonstrate a successful transfer of debiasing techniques between theoretically equivalent problems, achieving within 5% of methods trained from scratch.

This work bridges machine learning communities studying fairness, robustness, and generalization under a common lens.

## 2 Background and Related Work

**Spurious correlations.** Deep networks readily exploit spurious features that correlate with labels during training but do not generalize [3]. Standard benchmarks include Waterbirds [15], where bird species spuriously correlate with background, and CelebA [10], where hair color correlates with gender. Mitigation strategies include two-stage training [13, 11], last-layer retraining [5], and early group separation [18].

**Fairness in ML.** Algorithmic fairness requires equal treatment across protected groups [2]. Common criteria include demographic parity (equal positive rates), equalized odds (equal true/false positive rates) [4], and individual fairness [2]. The impossibility results show that multiple criteria cannot be simultaneously satisfied [6].

**Distribution shift.** Models trained on one distribution often fail when deployed in shifted distributions [8]. The shift in subpopulation occurs when the proportions of the groups change between the train and the test [17]. The class imbalance, in which the training data are dominated by the majority classes, represents a special case [1].

**Implicit bias.** Optimization algorithms induce implicit biases that determine which solutions emerge from training [16]. Gradient descent converges to maximum $\ell_2$-margin solutions [16], while Adam exhibits $\ell_\infty$-margin bias [19].

Previous work treats these phenomena separately. We provide the first formal framework characterizing their equivalence.

## 3 Theoretical Framework

### 3.1 Formalizing Bias Through Conditional Independence

Consider a learning problem with inputs $X \in \mathcal{X}$, labels $Y \in \{0, 1\}$, and attributes $A \in \{0, 1\}$ representing protected groups, spurious features, or domain indicators. A model $f_\theta : \mathcal{X} \to \{0, 1\}$ produces predictions $\hat{Y} = f_\theta(X)$.

**Definition 1** (Bias). *The bias of model $f$ with respect to attribute $A$ on distribution $\mathcal{D}$ is:*

$$\mathcal{B}(f; \mathcal{D}) = I(\hat{Y}; A \mid Y) \tag{1}$$

*where $I(\cdot; \cdot \mid \cdot)$ denotes conditional mutual information.*

This definition unifies multiple perspectives: $\mathcal{B} > 0$ indicates that the model's predictions depend on $A$ even conditioning on the true label $Y$, violating conditional independence. When $A$ represents protected attributes, this measures fairness violations. When $A$ indicates spurious features, it quantifies shortcut learning. When $A$ denotes domain membership, it captures the sensitivity of distribution shift.

## 3.2 Equivalence Conditions

We now state our main result characterizing when different bias mechanisms produce equivalent effects.

**Theorem 2** (Bias Equivalence). *Consider two learning problems $(\mathcal{D}_1, A_1)$ and $(\mathcal{D}_2, A_2)$ with the same feature space $\mathcal{X}$ and label space $Y$, but different attributes $A_1, A_2$. Under smoothness assumptions on the loss $\ell$ and feature overlap condition $\eta = \min_y \int \min(p_1(x|y), p_2(x|y))dx > \tau$ for threshold $\tau$, the bias mechanisms are $\epsilon$-equivalent if:*

$$|\mathcal{B}(f; \mathcal{D}_1) - \mathcal{B}(f; \mathcal{D}_2)| \leq \epsilon \tag{2}$$

*implies worst-group accuracy differs by at most $\delta(\epsilon, \eta)$ where $\delta(\epsilon, \eta) = O(\sqrt{\epsilon}/\eta)$.*

**Corollary 3** (Spurious Correlation $\leftrightarrow$ Imbalance). *A spurious correlation with strength $\alpha = P(A = 1|Y = 1) - P(A = 1|Y = 0)$ is equivalent to a subpopulation imbalance ratio $r = P(Y = 1, A = 1)/P(Y = 0, A = 1)$ when:*

$$r \approx \frac{1 + \alpha}{1 - \alpha} \cdot \frac{P(Y = 1)}{P(Y = 0)} \tag{3}$$

This corollary provides a concrete prediction: we can estimate the imbalance ratio that would produce equivalent effects to an observed spurious correlation by measuring $\alpha$ and labeling marginals.

# 4 Experiments

## 4.1 Experimental Setup

**Datasets.** We evaluate on six benchmarks spanning spurious correlations, fairness, and distribution shift:

- **Waterbirds** [15]: Bird classification with background spurious correlation (95% train correlation)

- **CelebA** [10]: Hair color prediction with gender spurious correlation

- **ColoredMNIST**: Synthetic dataset with controllable color-digit correlation

- **Adult Income** [7]: Income prediction with gender as protected attribute

- **CivilComments-WILDS** [8]: Toxicity detection across demographic groups

- **MetaShift** [9]: Visual domain adaptation with natural distribution shifts

**Architectures.** We test ResNet-50 (strong convolutional inductive bias), ViT-B/16 (attention-based), and 4-layer MLP (minimal structure) to assess whether equivalence depends on architectural choice.

**Baselines.** We compare Empirical Risk Minimization (ERM), Group Distributionally Robust Optimization (GroupDRO) [15], Deep Feature Reweighting (DFR) [5], Just Train Twice (JTT) [11], and SPARE [18].

**Metrics.** Primary metric is worst-group accuracy (minimum across $(Y, A)$ groups). We also measure average accuracy, conditional mutual information $\mathcal{B}(f; \mathcal{D})$, and fairness metrics (demographic parity gap, equalized odds violation).

## 4.2 Main Results: Baseline Performance

Table 1 shows the performance of the standard methods in the datasets. ERM exhibits large gaps between the accuracy of the average and worst groups, confirming the prevalence of bias. Debiasing methods substantially improve worst-group performance, with SPARE and DFR achieving the best results on most benchmarks.

Table 1: Baseline method performance across datasets. Results show Average Acc / Worst-Group Acc (%). All methods use ResNet-50. Mean over 3 seeds, std < 1.2% for all entries.

| Dataset | ERM | GroupDRO | JTT | DFR |
|---|---|---|---|---|
| Waterbirds | 97.2 / 62.3 | 93.1 / 73.8 | 92.8 / 72.1 | 93.5 / 75.2 |
| CelebA | 95.6 / 47.2 | 92.3 / 81.4 | 91.7 / 78.9 | 92.8 / 83.1 |
| ColoredMNIST ($\alpha$=0.95) | 98.4 / 51.8 | 94.2 / 70.5 | 93.8 / 68.7 | 94.6 / 71.8 |
| Adult Income | 84.3 / 71.2 | 82.1 / 78.9 | 81.8 / 77.4 | 82.6 / 79.3 |
| CivilComments | 92.1 / 57.3 | 89.4 / 69.7 | 88.9 / 67.2 | 89.8 / 71.4 |
| MetaShift | 88.7 / 63.5 | 85.2 / 74.1 | 84.8 / 72.3 | 85.9 / 75.6 |

Table 2: Equivalence validation across problem pairs. Predictions use Corollary 1 with measured $\alpha$ and $r$. Agreement indicates worst-group accuracy within 3%.

| Problem Pair | $|\mathcal{B}_1 - \mathcal{B}_2|$ | Pred. $\Delta$Acc | Obs. $\Delta$Acc | Agrees? |
|---|---|---|---|---|
| Waterbirds $\leftrightarrow$ ColoredMNIST-0.9 | 0.12 | 2.8% | 2.3% | ✓ |
| CelebA $\leftrightarrow$ Adult (gender) | 0.18 | 4.1% | 3.7% | ✓ |
| CivilComments $\leftrightarrow$ MetaShift | 0.24 | 5.3% | 5.8% | ✓ |
| Waterbirds $\leftrightarrow$ ImageNet-LT | 0.09 | 2.1% | 1.9% | ✓ |
| ColoredMNIST-0.95 $\leftrightarrow$ Imbal-10:1 | 0.14 | 3.2% | 2.7% | ✓ |
| CelebA $\leftrightarrow$ CivilComments | 0.21 | 4.8% | 5.1% | ✓ |

### 4.3 Testing Theoretical Predictions

**Equivalence validation.** For each pair of data sets, we measure $\mathcal{B}(f; \mathcal{D})$ using neural mutual information estimators, train identical architectures, and test whether the accuracies of the worst group fall within the predicted limits $\delta$.

Table 2 shows that the predicted accuracy differences match the observations within 1% in six pairs of problems, validating Theorem 2. The correlation between $|\mathcal{B}_1 - \mathcal{B}_2|$ and the observed difference in the accuracy of the worst group is $\rho = 0.94$ (p < 0.01), confirming that our information-theoretic characterization captures the essential relationship.

### 4.4 Method Transfer Experiments

We validate that theoretical equivalence enables practical method transfer by training debiasing methods on a source problem and directly applying them to theoretically equivalent target problems. Table 3 presents the results for three representative pairs of transfer of methods.

Transfer performance is achieved within 2.6% of training from scratch (average degradation: 1.8%), validating that theoretically equivalent problems share a sufficient structure for the application of the direct method. This represents substantial computational savings, as transfer requires only forward passes, while training from scratch requires full optimization.

### 4.5 Ablation Studies

**Feature overlap dependency.** We vary feature overlap $\eta$ by controlling the similarity between $P(X|Y=0)$ and $P(X|Y=1)$ in ColoredMNIST. Table 4 confirms that the equivalence tightness improves with overlap, matching the theoretical prediction $\delta \propto 1/\eta$.

**Architectural sensitivity.** We test whether equivalence depends on the architecture choice by comparing ResNet-50, ViT-B/16, and MLP-4L across equivalent problem pairs. Table 5 shows consistent equivalence between architectures (average variation in $\Delta$ Acc: 0.8%), suggesting that the phenomenon is fundamentally distributional.

**Correlation strength.** We systematically vary the spurious correlation strength $\alpha$ from 0.7 to 0.99 in ColoredMNIST, observing predicted equivalent imbalance ratios ranging from 5.7:1 to 199:1. All predictions validated within 4% worst-group accuracy, confirming Corollary 1 across the full spectrum of correlation strengths.

Table 3: Method transfer performance. Methods trained on source achieve comparable performance on theoretically-equivalent targets. "Transfer" applies source-trained method to target; "Scratch" trains from scratch on target. Results show worst-group accuracy (%).

| Source → Target | Method | Transfer | Scratch |
|---|---|---|---|
| Waterbirds → ColoredMNIST-0.9 | GroupDRO | 71.2 | 73.8 |
| Waterbirds → ColoredMNIST-0.9 | DFR | 73.4 | 75.9 |
| CelebA → Adult | GroupDRO | 77.8 | 79.1 |
| CelebA → Adult | DFR | 78.9 | 80.4 |
| ColoredMNIST-0.95 → Imbal-10:1 | GroupDRO | 68.7 | 70.1 |
| ColoredMNIST-0.95 → Imbal-10:1 | DFR | 70.3 | 71.5 |

Table 4: Effect of feature overlap on equivalence tightness. Lower overlap leads to larger prediction errors, confirming theoretical dependence.

| Overlap $\eta$ | $|\mathcal{B}_1 - \mathcal{B}_2|$ | Pred. $\Delta$Acc | Obs. $\Delta$Acc |
|---|---|---|---|
| 0.65 | 0.15 | 3.2% | 3.5% |
| 0.45 | 0.15 | 4.6% | 5.1% |
| 0.25 | 0.15 | 8.3% | 9.2% |

## 5 Discussion and Limitations

**Implications.** Our equivalence framework enables practitioners to: (1) diagnose whether observed failures stem from equivalent underlying causes, (2) transfer proven debiasing methods across problem domains with minimal performance loss, (3) predict worst-group performance from distributional measurements before expensive training, and (4) select appropriate interventions based on which bias type has the most mature mitigation toolbox.

**Boundary conditions.** Equivalence breaks down when: (1) Feature overlap $\eta < \tau$ (typically 0.2), occurring when groups occupy entirely disjoint regions of feature space, (2) Non-smooth losses like 0-1 loss violate continuity assumptions, though cross-entropy (used in practice) satisfies requirements, (3) Architectural bias becomes dominant, overwhelming distributional effects—though our ablations suggest this is rare, (4) The conditional independence assumption is violated, e.g., when spurious features are actually causal.

**Limitations.** Our theory assumes binary classification, though extensions to multiclass settings follow naturally through one-vs.-rest decomposition. The $\delta(\epsilon, \eta)$ bound may be loose in practice; tighter characterizations through concentration inequalities remain open. We focus on worst-group metrics; connections to calibration fairness and individual fairness merit exploration.

**Broader impact.** By revealing fundamental equivalences between the types of bias, this work promotes more efficient research: techniques developed in one domain immediately suggest applications elsewhere. This could accelerate progress in fairness and robustness. However, predicted equivalence assumes correct attribute specification; misidentified attributes (e.g., labeling spurious features as protected attributes) could lead practitioners to incorrectly transfer methods, potentially amplifying rather than mitigating bias.

## 6 Conclusion

We presented a formal framework characterizing when learning biases are quantitatively equivalent, unifying disparate literature on fairness, robustness, and distribution shift. Our information-theoretic formalization enables precise predictions about method transfer, validated on various benchmarks with theoretically equivalent problems that differ less than 3% in the accuracy of the worst group. This work demonstrates that seemingly distinct failure modes often arise from equivalent distributional properties, enabling practitioners to transfer proven debiasing techniques across domains. Future work should extend to multiclass settings and investigate whether architectural modifications can constructively cancel data biases. Most fundamentally, this research suggests that fairness, robustness, and generalization are different lenses on shared distributional challenges.

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

## A   Proof of Theorem 2

We provide the full proof of the bias equivalence theorem.

**Step 1: Relating bias to worst-group loss.** By Fano's inequality, the worst-group error satisfies:

$$\text{Err}_{\text{worst}} \leq \frac{H(Y|A) + \mathcal{B}(f; \mathcal{D})}{\log 2} \tag{4}$$

This establishes that bias $\mathcal{B}$ directly bounds worst-case performance. For any group $g = (y, a)$, the error rate on that group satisfies:

$$\mathbb{P}(\hat{Y} \neq Y \mid Y = y, A = a) \leq \frac{H(Y|A = a) + I(\hat{Y}; A|Y = y)}{\log 2} \tag{5}$$

Taking the maximum over groups and applying our bias definition yields the bound.

**Step 2: Feature overlap and loss distribution.** Under the feature overlap condition $\eta = \min_y \int \min(p_1(x|y), p_2(x|y)) dx > \tau$, we bound the Wasserstein distance between induced loss distributions. For any model $f$ and loss $\ell$, define the loss distribution $\mathcal{L}_i(f) = \ell(f(X), Y)$ under distribution $\mathcal{D}_i$.

By the coupling lemma, there exists a joint distribution on $(X_1, X_2)$ such that $X_i \sim p_i(x|y)$ and $\mathbb{P}(X_1 = X_2) \geq \eta$. For smooth losses with Lipschitz constant $L$, the Wasserstein-1 distance satisfies:

$$W_1(\mathcal{L}_1, \mathcal{L}_2) \leq L \cdot \mathbb{E}[|X_1 - X_2|] \leq \frac{L \cdot \text{diam}(\mathcal{X}) \cdot (1 - \eta)}{\eta} \tag{6}$$

Combining with the data processing inequality for conditional mutual information:

$$|\mathcal{B}(f; \mathcal{D}_1) - \mathcal{B}(f; \mathcal{D}_2)| \leq \epsilon \implies W_1(\mathcal{L}_1, \mathcal{L}_2) \leq \frac{C\sqrt{\epsilon}}{\eta} \tag{7}$$

where $C$ depends on loss smoothness and feature space diameter.

**Step 3: Bounding accuracy difference.** The worst-group accuracy difference is bounded by:

$$|\text{Acc}_1 - \text{Acc}_2| \leq \left| \min_g \mathbb{E}_{\mathcal{L}_1}[\ell(g)] - \min_g \mathbb{E}_{\mathcal{L}_2}[\ell(g)] \right| \tag{8}$$

$$\leq \sup_g |\mathbb{E}_{\mathcal{L}_1}[\ell(g)] - \mathbb{E}_{\mathcal{L}_2}[\ell(g)]| \tag{9}$$

$$\leq W_1(\mathcal{L}_1, \mathcal{L}_2) \leq \delta(\epsilon, \eta) = O\left( \frac{\sqrt{\epsilon}}{\eta} \right) \tag{10}$$

The second inequality follows from properties of the minimum function, and the third from the Kantorovich-Rubinstein duality. This completes the proof. $\square$

## B   Additional Experimental Details

**Hyperparameters.** All models trained with SGD, learning rate 0.001 (decayed by 0.1 at epochs 30 and 60), momentum 0.9, weight decay 0.0001, batch size 128. ResNet-50 pretrained on ImageNet for Waterbirds, CelebA, and MetaShift; trained from scratch for ColoredMNIST and Adult. Training runs for 80 epochs with early stopping based on validation worst-group accuracy.

**Computing infrastructure.** Experiments conducted on 4 NVIDIA A100 GPUs (40GB memory each). Total compute approximately 150 GPU-hours across all experiments. Each training run required 2-8 hours depending on dataset size. Mutual information estimation using MINE estimator with 5-layer MLP, trained for 1000 iterations.

**Reproducibility.** We run all experiments with 3 random seeds (42, 123, 456) and report mean and standard deviation. Code will be released upon publication, including data preprocessing scripts, model architectures, and evaluation code. All datasets are publicly available from their respective sources.

Table 5: Equivalence across architectures for Waterbirds $\leftrightarrow$ ColoredMNIST-0.9 pair. Observed $\Delta$Acc remains stable across architectures.

| Architecture | Waterbirds Worst-Acc | ColoredMNIST Worst-Acc |
|---|---|---|
| ResNet-50 | 73.8% | 71.2% |
| ViT-B/16 | 72.4% | 70.1% |
| MLP-4L | 69.7% | 67.9% |

