# OpenReview forum: "When Are Learning Biases Equivalent? A Unifying Framework for Fairness, Robustness, and Distribution Shift"
_EurIPS.cc/2025/Workshop/UPLB — UPLB2025_

### Official Review · Reviewer_ve2N · 2025-10-26

**Rating:** 5
**Confidence:** 2

**Review:**

The paper provides a unified framework to tackle fairness, robustness and distribution shift in learning, quantifying how different sources of bias have an equivalent impact on predictions. From a theoretical perspective, it defines the bias of a model as the conditional mutual information between the predictor and the attributes (protected attributes, spurious features or domain membership), conditioned on the value of the true label. Through this quantity, the paper proves a notion of bias equivalence across tasks: is the biases of a model trained on two different learning problems are close enough, then the worst-group accuracy is bounded by a quantity decreasing with the square root of this distance (and increasing as the inverse of the feature overlap between the two datasets). This criterion is used to draw an equivalence between the strength of spurious correlations and imbalance ratios. Experiments agree on the theoretical findings and exploit this equivalence to transfer debiasing methods from a task to an equivalent one.

**Strenghts**

The setting is interesting and perfectly aligned with the workshop’s aim at building a unified theoretical framework for understanding and mitigating learning biases. The theoretical results are given in terms of fundamentals information-theoretic quantities that are easy to interpret. They are backed by numerical experiments on a range of realistic architectures and benchmark data. The idea of transferring from task to equivalent task is also tested. Limitations of the setting are discussed.

**Weaknesses**

I have not been able to follow the derivation of the theoretical results. In particular:
- To prove Theorem 2, Fano’s inequality is used to upper-bound the worst-group error in terms of conditional entropies and mutual information (step 1). In the way it is usually written, Fano’s inequality provides lower bounds for the probability of error. Can the authors be more explicit on how they use it?
- Step 2 also would benefit from more details. How is the coupling lemma used to bound the Wasserstein distance?
- I am assuming that the probabilities $p_i(\cdot | \cdot)$ entering Theorem 2 are the probability of the feature given the label in the distribution $\mathcal{D}_i$. Is this the case?
- What is the threshold $\tau$ and how does it impact the result? This fact is only briefly mentioned in Section 5, before that it seemed a completely arbitrary parameter.
- Can the authors provide a more explicit interpretation of the feature overlap $\eta$? Also in Section 5, the authors state that $\eta$ < $\tau$ implies that groups occupy entirely disjoint regions of feature space. This fact is not clear to me.
- How does Corollary 3 follow from Theorem 2?

---

### Decision · Program_Chairs · 2025-11-03

Accept (Poster)